# Down-Regulation of Double C2 Domain Alpha Promotes the Formation of Hyperplastic Nerve Fibers in Aganglionic Segments of Hirschsprung’s Disease

**DOI:** 10.3390/ijms231810204

**Published:** 2022-09-06

**Authors:** Jun Xiao, Xinyao Meng, Ke Chen, Jing Wang, Luyao Wu, Yingjian Chen, Xiaosi Yu, Jiexiong Feng, Zhi Li

**Affiliations:** 1Department of Pediatric Surgery, Tongji Hospital, Huazhong University of Science and Technology, Wuhan 430030, China; 2Hubei Clinical Center of Hirschsprung’s Disease and Allied Disorders, Wuhan 430030, China; 3Department of Pediatric Surgery, Fujian Maternity and Child Health Hospital, Affiliated Hospital of Fujian Medical University, Fuzhou 350001, China

**Keywords:** Hirschsprung disease, hyperplastic nerve fibers, DOC2A, zebrafish, neural sphere

## Abstract

Hirschsprung’s disease (HSCR) is a common developmental anomaly of the gastrointestinal tract in children. The most significant characteristics of aganglionic segments in HSCR are hyperplastic extrinsic nerve fibers and the absence of endogenous ganglion plexus. Double C2 domain alpha (DOC2A) is mainly located in the nucleus and is involved in Ca^2+^-dependent neurotransmitter release. The loss function of DOC2A influences postsynaptic protein synthesis, dendrite morphology, postsynaptic receptor density and synaptic plasticity. It is still unknown why hyperplastic extrinsic nerve fibers grow into aganglionic segments in HSCR. We detected the expression of DOC2A in HSCR aganglionic segment colons and established three DOC2A-knockdown models in the Neuro-2a cell line, neural spheres and zebrafish separately. First, we detected the protein and mRNA expression of DOC2A and found that DOC2A was negatively correlated with AChE^+^ grades. Second, in the Neuro-2a cell lines, we found that the amount of neurite outgrowth and mean area per cell were significantly increased, which suggested that the inhibition of DOC2A promotes nerve fiber formation and the neuron’s polarity. In the neural spheres, we found that the DOC2A knockdown was manifested by a more obvious connection of nerve fibers in neural spheres. Then, we knocked down Doc2a in zebrafish and found that the down-regulation of Doc2a accelerates the formation of hyperplastic nerve fibers in aganglionic segments in zebrafish. Finally, we detected the expression of MUNC13-2 (UNC13B), which was obviously up-regulated in Grade3/4 (lower DOC2A expression) compared with Grade1/2 (higher DOC2A expression) in the circular muscle layer and longitudinal muscle layer. The expression of UNC13B was up-regulated with the knocking down of DOC2A, and there were protein interactions between DOC2A and UNC13B. The down-regulation of DOC2A may be an important factor leading to hyperplastic nerve fibers in aganglionic segments of HSCR. UNC13B seems to be a downstream molecule to DOC2A, which may participate in the spasm of aganglionic segments of HSCR patient colons.

## 1. Introduction

Hirschsprung disease (HSCR) is one of the most common developmental anomalies of gastrointestinal tracts in children, and the generally cited prevalence is 1 in 5000 live births [1]. The typical clinical manifestations are intractable constipation and gastrointestinal tract obstruction, and severe cases manifested as obvious abdominal distension and malnutrition [2]. The most significant characteristic of aganglionic segments in HSCR lacks an endogenous ganglion plexus (myenteric and submucous plexuses). Meanwhile, the hyperplastic exogenous nerves (from pelvic autonomic ganglion cells, spinal sensory ganglia and paraspinal sympathetic ganglia [3]), especially the branches of the cholinergic autonomic nerve, course through the bowel wall [3,4,5,6], and extrinsic preganglionic parasympathetic fibers markedly increase [7,8].

Double C2 domain alpha (DOC2A) is mainly located in the nucleus and is suggested to be involved in Ca^2+^-dependent neurotransmitter release [9]. Furthermore, DOC2A was related to the development of neurological diseases, including autism and epilepsy [10,11,12]. The loss function of DOC2A augmented glutamatergic synaptic strength in spontaneous neurotransmission and decreased spontaneous neurotransmitter releasing [13,14], which influenced postsynaptic protein synthesis, dendrite morphology, postsynaptic receptor density and synaptic plasticity [15,16,17]. Our previous study also validated that the knockdown of DOC2A enhanced mouse hippocampal neuron polarity [18]. It is still unknown why nerve fibers grow into aganglionic segments in HSCR. The purpose of this study was to analyze the potential mechanism of DOC2A in the hyperplastic nerve fibers in aganglionic segments of HSCR.

## 2. Results

### 2.1. Expression of DOC2A Was Correlated with AChE^+^ Grades in HSCR

A characteristic of HSCR is the absence of enteric neurons in the distal colon, along with the hyperplastic exogenous nerves coursing through the bowel wall (Figure 1).

Aganglionic segments of HSCR patients and model mice were divided into four grades according to AChE (acetylcholinesterase) positive (AChE^+^) nerves. The AChE immunohistochemistry of the distal aganglionic colon was collected to identify mucosal innervation grades (Figure 2). Based on the mucosal innervation grades, we then detected the protein and mRNA expression of DOC2A and found that the DOC2A was negatively correlated with AChE^+^ grades (Figure 3).

### 2.2. Knock-Down of DOC2A Promoted Nerve Fiber Formation In Vitro

In order to confirm the correlation between DOC2A and nerve fiber generation, we used an shRNA lentivirus transfection technique to knock down the expression of DOC2A in Neuro-2a (N-2a) cell lines (Appendix A) and found that the amount of neurite outgrowth and the mean area per cell were significantly increased, which suggested that the inhibition of DOC2A promotes the nerve fiber formation and the neuron’s polarity (Figure 4).

To further study the nerve fiber formation in enteric neural cells, we then isolated neural spheres from the colons of 13.5-day-old (E13.5d) embryonic mice. The shRNA lentivirus transfection technique was used to knock down the expression of DOC2A (Appendix A). We found that the DOC2A knockdown was manifested by a more obvious connection of nerve fibers in neural spheres (Figure 5).

### 2.3. Down-Regulation of Doc2a Accelerated the Hyperplastic Nerve Fibers’ Formation in Aganglionic Segment in Zebrafish

First, we constructed a Doc2a knockdown model in zebrafish by Morpholino (MO). However, the number of enteric neurons and the morphology of nerve fibers were not significantly affected by the loss of Doc2a compared with the wild type (Figure 6A,B,E,F,J). Then, we knocked down the ret gene in zebrafish and observed an obvious phenotype of aganglionosis on the fifth day, while there were no obvious hyperplastic nerve fibers at this time (Figure 6C,G). Further, we knocked down Doc2a and Ret in zebrafish simultaneously. The double knockdown of Ret and Doc2a showed a visible nerve fiber formation as compared with the *ret*-MO group and control group (*ret*-MO group and *ret*-mismatch-MO group show similar phenotypes) (Figure 6C,D,K), and there were no enteric neurons in both the double knockdown group and *ret*-MO group (Figure 6G,H,J).

In conclusion, the down-regulation of Doc2a can accelerate hyperplastic nerve fiber formation in aganglionic segments in zebrafish.

### 2.4. UNC13B Seems to Be a Downstream Molecule to DOC2A

There are strong interconnections between DOC2 and MUNC-13 proteins by phorbol ester through the C1 domain of Munc-13 [19]. At present, studies on DOC2A have mainly been based on the hippocampal neurons of the brain, and DOC2A was mainly used to interact with MUNC-13 (UNC13), which had positive effects on the release of neurotransmitters [19]. DOC2A and UNC13 proteins were involved in Ca^2+^-dependent neurotransmitter release as they were located in synaptic vesicles and the presynaptic plasma membrane, respectively [20].

Hence, we detected the expression of MUNC13-1 (UNC13A) and MUNC13-2 (UNC13B) in DOC2A knockdown cells (N-2a cell lines and neural spheres). Interestingly, we found that UNC13A was down-regulated while UNC13B was up-regulated (Figure 7A,B). UNC13B was significantly up-regulated in Grade3/4 (lower DOC2A expression) compared to Grade1/2 (higher DOC2A expression) in the circular muscle layer and longitudinal muscle layer (Figure 7C–J). We also confirmed that there are protein interactions between DOC2A and UNC13B by co-immunoprecipitation (CO-IP) (Figure 7K,L). UNC13B seems to be a downstream molecule to DOC2A, which may participate in the spasm of aganglionic segments of HSCR patient colons. However, the signaling pathway mechanisms require further research.

## 3. Discussion

HSCR is characterized by a lack of neuron cells and a marked increase of hyperplastic exogenous nerve fibers. However, it is still unclear why the hyperplastic exogenous nerves course through the bowel wall and cause the spasm of aganglionic segments in the colon of HSCR patients.

In the present study, we found that the expression of DOC2A in aganglionic segments was correlated with AChE^+^ grades. We established three DOC2A-knockdown models in the N-2a cell line, neural spheres and zebrafish separately, and all of them showed nerve fiber disorder. It seemed that the down-regulation of DOC2A can increase the number of nerve fiber connections, which is similar to the phenomenon in which parasympathetic and sympathetic fibers proliferate largely and distribute intricately in aganglionic segments of HSCR. However, knocking down Doc2a in zebrafish did not result in a loss of intestinal neurons or hyperplastic nerve fibers as we injected different concentration gradients of MO, which was different from the phenotypes of cell experiments. We speculated that zebrafish might have a strong compensatory mechanism in response to the down-regulation of Doc2a, and the function of Doc2a in the intestinal tract is not significant. Interestingly, when we constructed a model of the absence of enteric neurons caused by *ret*-MO, we observed that the down-regulation of Doc2a can accelerate hyperplastic nerve fibers in aganglionic segments in zebrafish. This is because the exogenous hyperplastic nerves, which are from the sympathetic ganglion, especially the branches of the cholinergic autonomic nerve, course through the bowel wall and extend inside the intestinal tract [3,4,5,6]. Therefore, we doubt that Doc2a influences the polarity of the sympathetic neurons and promotes more nerve fibers connections.

The abnormal release of neurotransmitters is the main factor of spasm, which usually appears in aganglionic segments of HSCR. The neurotransmitters may indirectly affect the receptors on the interstitial cells of Cajal (ICCs) cells and affect the smooth muscle contraction or directly affect the transmitter receptors on the smooth muscle [21]. DOC2A and UNC13 proteins were involved in Ca^2+^-dependent neurotransmitter release as they were located in synaptic vesicles and the presynaptic plasma membrane, respectively [20]. In our study, we found that UNC13A was down-regulated while UNC13B was up-regulated, and we also confirmed that there are protein interactions between DOC2A and UNC13B by co-immunoprecipitation (Figure 7K,L). UNC13B seems to be a downstream molecule to DOC2A, which may participate in the spasm of aganglionic segments of HSCR patient colons. However, the signaling pathway mechanisms require further research.

As DOC2A is involved in Ca^2+^-dependent neurotransmitter release [9] and UNC13 is a key factor in clear synaptic vesicle maturation, regulation by Ca2^+^ may be a way to treat hyperplastic nerve fibers of HSCR. This study provides new insights into the formation of hyperplastic nerve fiber formation, which is essential for precise treatment and prevention of postoperative recurrence of HSCR.

As a neural crest-derived mouse cell line, N-2a cells have been extensively utilized to research neuronal development, axonal growth, and signaling pathways [22] because of large amounts of tubulin, which acts as a contractile system in nerve cells in response to axoplasmic flow. Enteric neurons in the hindgut are derived from separate vagal and sacral neural crest cells [23,24]. Neural spheres extracted from colons of E13.5d mice mainly contain neural crest cells (NCCs) (Appendix A) and enteric mesenchymal cells (EMCs) [25,26], while reciprocal induction of EMCs and NCCs promotes the development of the gut [27]. One study also showed that neural spheres could improve their function after being transplanted into an aganglionic embryonic distal colon [28]. As DOC2A may influence the polarity of sympathetic neurons and hyperplastic nerve fibers grow into the distal colon, neural spheres seem to be a good model to simulate the biological environment of the colon, and a zebrafish is a good biological model for studying the development of intestinal neurons [29]. Based on the backgrounds above, we chose these three models to carry out our study.

## 4. Materials and Methods

### 4.1. Approval of the Study and Experimental Specimens

From 2019 to 2020, pathological specimens were acquired from HSCR patients who underwent surgery at the Tongji Hospital of Huazhong University of Science and Technology. Aganglionic segments were chosen based on the colon’s pathologic reports. Their paired normal tissues came from patients who had been injured in road accidents (Appendix A). Before using clinical materials, all patients (or their legal guardian) signed informed consent forms. The ethical committee of Tongji Hospital with Huazhong University of Science and Technology approved the use of tissues for this study (Ethical approval number: 2020-S226). The Jackson Laboratory provided a breeding colony of Ednrbtm1Ywa/J heterozygous mice (Ednrb^tm1Ywa/J^ on a hybrid C57BL/6J-129Sv background) (JAX-003295). In 2-week-old Ednrb^−/−^ mice with an evident HSCR phenotype, we separated the aganglionic segments (Appendix A). The Institutional Animal Care and Use Committee at Tongji Hospital approved the animal study protocols (Ethical approval number: TJH-202111024).

### 4.2. Histopathology of the Colon

Hematoxylin and eosin (HE) stains were used on paraffin-embedded colon sections (Appendix A).

### 4.3. Nerve Fiber Scoring

Nerve fiber innervation was classified into four grades based on the quantity of AChE^+^ nerve fibers: grade 1 (absent innervation), grade 2 (poor innervation), grade 3 (intermediated innervation), and grade 4 (complete innervation) (high innervation) (Figure 2A–J). Images were examined using a brightfield microscope and CellSens Dimension 2.2 Software (Olympus) to quantify the AChE^+^ nerve fibers in the mucosal areas. Six clipped regions of interest containing 20 epithelial crypts from the distal colon were investigated on average, resulting in 60 crypts per patient/mice. According to the visual scoring, a total of 12 patients/mice (3 patients/mice per innervation grade) were chosen. On each cropped image, a manual threshold was applied, and item counts were calculated using the Olympus CellSens dimension program.

### 4.4. Culture of Cells

OBIO (OBIO Technology Corp, Shanghai, China) provided the N-2a cell lines, which were grown at 37 °C in a humidified incubator in minimum Eagle’s (MEM) medium with 10% fetal bovine serum (Gibco, Grand Island, NY, USA). Neural spheres were isolated from E13.5d C57BL/6 mice colon and cultured at 37 °C. Neural spheres were grown in DMEM/F-12 media with 1 percent N_2_ supplement (Life Technologies, Carlsbad, CA, USA), 1 percent B27 supplement (Life Technologies, Carlsbad, CA, USA), 10 ng/mL fibroblast growth factor (FGF) (Life Technologies, Carlsbad, CA, USA), 1 percent Glutamate, and 1 percent penicillin–streptomycin.

### 4.5. Transfection

Inoculated cells: The N-2a cells were inoculated in 6-well plates, and the cell density was controlled at 50%. The extracted primary cells were cultured in 6-well plates. On the third day, a lot of suspended neural spheres were formed. The neural spheres were cultured in specific medium (adding enough medium) described in Section 4.4. On the third day, we used the lentivirus to knock down the expression of DOC2A. The lentivirus was used to knock down the expression of DOC2A (pLKD-CMV-Puro-U6-shRNA(*Doc2a*), Obio Technology Corp, Shanghai, China). We set three groups in the N-2a cell line and neural spheres separately, including infected with *Doc2a*-lentivirus (knockdown group), Lentivirus Luciferase Reporter negative control (negative control group), and cultured with an equal medium (blank control group). Calculating virus load: (MOI) × (number of cells)/(virus drops degree). N-2a, MOI = 10; Neural sphere, MOI = 80. Transfection progress was carried out according to the instructions using the Lipofectamine 2000 kit (Invitrogen, Carlsbad, CA, USA). Puromycin (2 g/mL) was used for four days to remove untransfected cells. The cells were collected and prepared for the next experiments of Western blot and quantitative real-time PCR.

### 4.6. Quantitative Real-Time PCR (qRT-PCR)

Total RNA was extracted from cells by Trizol reagent (Servicebio, Wuhan, China). HiScript II (Vazyme, Nanjing, China) was used to make cDNA. The reaction system of obtaining cDNA (20 uL): RNA (1 uL)+ 5 * HiScript II qRT SuperMix (4 uL) + RNase-free ddH_2_O (15 uL). All the procedures referred to the instruction of HiScript II (Vazyme, Nanjing, China). Reverse transcription conditions: 50 °C  for 15 min → 85 °C  for 5 s → 10 °C. PCR primers: β-ACTIN (Human): F-CCTTCCTGGGCATGGAGTC, R-TGATCTTCATTGTGCTGGGTG; β-Actin (mouse): F-TGCTGTCCCTGTATGCCTCTG, R-TGATGTCACGCACGATTTCC; *DOC2A* (Human): F-AATCCCGTGTGGAATGAGGAC, R-ACGCGGATCTCCCCAATAAAC; *Doc2a* (Mouse): F-GACATCACCCACAAGGTGCT, R-GGCGCTCAAGGCAGATGTTA; *Unc13a* (mouse): F-CAAGTTTGACGGTGCCCAAG, R-TGGAAGGAGTACTCATCCTGGT; *Unc13b* (mouse): F-CTGGACCTATTTGGGCTGGG, R-ACTACTGCTGGTAGGGCTGA. The reaction system (20 uL): 2 * ChamQ Universal SYBR qPCR Master Mix (10 uL) + PrimerR (10 uM, 0.5 uL) + PrimerF (10 uM, 0.5 uL) + Template DNA (1 uL) + DEPC Water (8 uL). Quantitative real-time PCR (qRT-PCR) for mRNA was carried out on Applied Biosystems StepOne Plus Real-Time PCR (Roche, Pleasanton, CA, USA). The results were exported and analyzed.

### 4.7. Western Blot Analysis

RIPA buffer (Sigma, Baton Rouge, LA, USA), including protease inhibitors, was used to lyse whole cellular proteins. Bicinchoninic acid (BCA) analysis was used to quantify the protein extractions (Beyotime, Shanghai, China). Protein extractions were separated on a 10% SDS-PAGE gel. Membranes were then incubated with peroxidase (HRP)-conjugated secondary antibody after being incubated with a high-affinity antibody (DOC2A (R&D SYSTEMS, MN, USA), UNC13B (ABclonal, Wuhan, China), β-actin (Proteintech, Wuhan, China). A chemiluminescence system (Bio-Rad, Hercules, CA, USA) and Image Lab Software were used to detect and analyze the signals.

### 4.8. Immunofluorescence (IF) Staining of N-2a Cells and Neural Spheres

After successfully knocking down DOC2A, we prepared for the experiment of IF. The different groups of N-2a cells and neural spheres were cultured in 24-well plates. As the N-2a cells showed adherent growth, we discarded the medium and washed the cells with PBS. We cultured the neural spheres with less volume of specific medium (as described in Section 4.4). At this time, we added new specific medium regularly rather than replacing the medium. The neural spheres would show adherent differentiation gradually. We set up three groups in the N-2a cell line and neural spheres, separately, including infected with *Doc2a*-lentivirus (knockdown group), Lentivirus Luciferase Reporter negative control (negative control group), and cultured with an equal medium (blank control group). We kept the culturing conditions and methods of each group consistent. Adherent cell morphology was fixed with 4% paraformaldehyde. Nonspecific antigens were blocked with BSA and washed carefully. Immunofluorescence labeling with the following antibodies was used to confirm the cultivated neural spheres: Nestin (Abcam, Cambridge, MA, USA), P75 (Abcam, Cambridge, MA, USA); P75 and Tuj1 (Abcam, Cambridge, MA, USA) were used for IF staining of neurite of neural spheres; N-2a cells were stained with Tuj1. To combine with appropriate antibodies, DyLight 488 Conjugated AffiniPure Goat Anti-mouse (BOSTER, Wuhan, China) and DyLight 594 Conjugated AffiniPure Goat Anti-rabbit (BOSTER, Wuhan, China) were utilized. A LSM 800 confocal microscope (Zeiss, Jena, Germany) was used to capture the images (Appendix A).

### 4.9. Morpholino (MO) Knockdown of doc2a and ret Genes

Morpholino antisense oligonucleotides and mismatch morpholino targeting the *doc2a* and *ret* gene were designed and synthesized by Genetools [30]. Mismatch morpholino can be seen as the negative control. The morpholinos’ sequences are as follows: 

*doc2a*, 5′-TTGTGGCTAAATGCACTCACTTCTT-3′;

*ret*, 5′-GTCAATCATAAGTGTTAATGTCACAA3′.

The optimum injection concentrations of *doc2a* and *ret* were 2 ng/uL, separately. Then we set three groups to investigate the phenotypes of double MO (*doc2a* and *ret*) injection, which were the *ret*-MO group, *ret*-mismatch-MO group and *ret-doc2a*-MO group.

In order to investigate the optimal injection concentration of *ret*-MO and *doc2a*-MO, we set up different concentration gradients, including 1 ng/uL, 2 ng/uL, 4 ng/uL and 6 ng/uL. After injection, we counted the mortality and malformation rates of zebrafish at different growth time points (0.2 hpf, 10 hpf, 1 d, 2 d, 3 d, 5 d and 7 d). We analyzed and concluded the results. The malformation rate and mortality were low, and the phenotype of the enteric neurons was the most stable when the injection concentration of *ret*-MO and *doc2a*-MO was 2 ng/uL (Appendix A).

### 4.10. Whole-Mount Immunofluorescence Staining

On the fifth day after injecting MO into zebrafish embryos, we collected 30 zebrafish from each group. Anti-HuC/D antibody (Life Technologies, Carlsbad, CA, USA) and anti-tubulin antibody were used for whole-mount immunofluorescent labeling (ImmunoStar, Hudson, WI, USA). Embryos were preserved overnight with 4% paraformaldehyde and then washed three times in phosphate buffered solution (PBS). After 1 h incubation in blocking solution (2 percent goat serum, 2 mg/mL bovine serum albumin (BSA) in 1 × PBS), embryos were treated with the primary antibody in blocking solution overnight at 4 °C. Then, embryos were incubated in a secondary antibody of Alexa Fluor rabbit anti-mouse IgG (Life Technologies, Carlsbad, CA, USA). Finally, an LSM 800 confocal microscope (Zeiss, Jena, Germany) was used to capture the images. Image J was then used to calculate the fluorescence intensity or number of tagged cells. According to the anatomical morphology of the zebrafish intestinal tract, we divided the intestinal tract of zebrafish into three zones, named zone 1, zone 2 and zone 3, respectively. Here, we observed and counted the number of neurons and the morphology of nerve fibers in zones 1 and 2. As for the analysis of hyperplastic nerve fibers in *ret*-MO zebrafish and *ret-doc2a*-MO, we measured the length of the intestinal tract containing hyperplastic nerve fibers, and then we calculated and compared the length proportion of the intestinal tract containing hyperplastic nerve fibers in zones 1 and 2.

### 4.11. Statistics

All results were given as means and standard deviations. Two-tailed Student’s *t*-tests were performed to analyze the differences between the two groups, with *p* < 0.05 regarded as statistically significant. As for the three groups, we performed one-way ANOVA with Bonferroni post hoc tests. All tests were carried out in triplicate and were repeated at least three times.

## Figures and Tables

**Figure 1 ijms-23-10204-f001:**
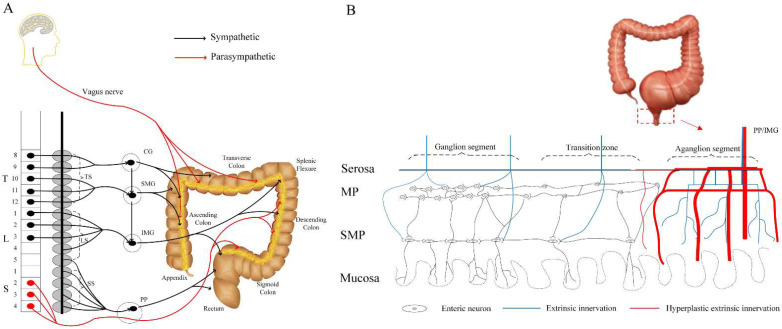
Intestinal innervation in the HSCR colon. (**A**) Intestinal innervation network. (**B**) Intrinsic nervous system and exogenous nerve fibers in the HSCR colon. Abbreviations: HSCR, Hirschsprung’s disease; T, thoracic vertebrae; L, lumbar vertebrae; S, sacral vertebrae; TS, thoracic spinal; LS, lumbar spinal; SS, sacral spinal; CG, celiac ganglion; SMG, superior mesenteric ganglion; IMG, inferior mesenteric ganglion; PP, pelvic nerve; ENS, enteric nervous system; MP, myenteric plexus; SMP, submucosal plexus.

**Figure 2 ijms-23-10204-f002:**
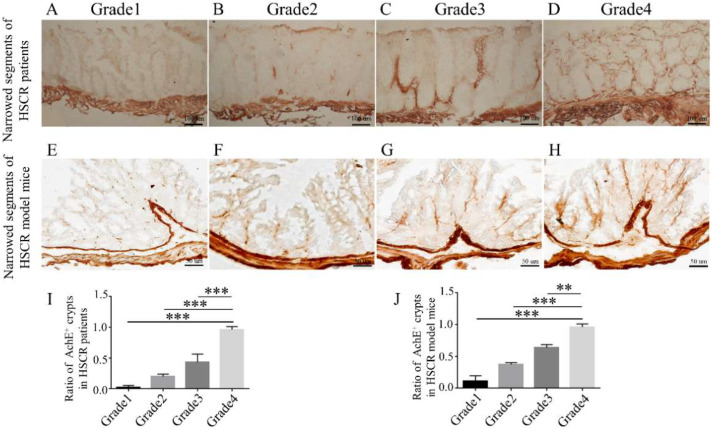
Aganglionic segments of HSCR patients and model mice were divided into four grades according to AChE^+^ nerves. (**A**–**D**) Different grades of AChE^+^ in colonic aganglionic segments of HSCR patients. (**E**–**H**) Different grades of AChE^+^ in colonic aganglionic segments of HSCR model mice. (**I**,**J**) The ratio of AchE^+^ crypts in the colons of HSCR patients and model mice. **, *p* < 0.01; ***, *p* < 0.001, one-way ANOVA with Bonferroni post hoc, *n* = 10.

**Figure 3 ijms-23-10204-f003:**
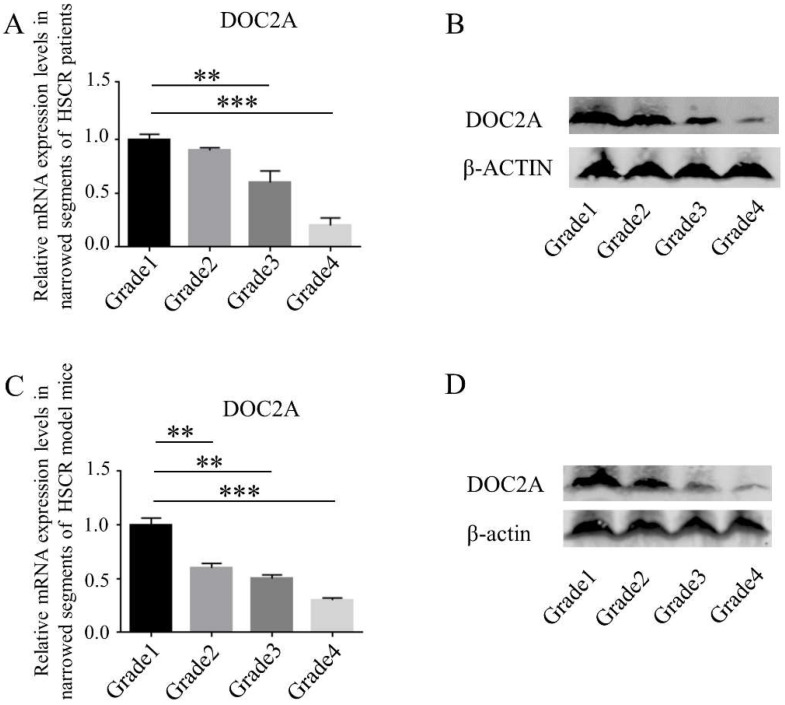
Relative expression levels of DOC2A in different grades. (**A**,**C**) Quantification of mRNA expression levels of DOC2A in different grades determined by RT-qPCR. (**B**,**D**) Protein expression of the DOC2A in different grades determined by western blots. **, *p* < 0.01; ***, *p* < 0.001, one-way ANOVA with Bonferroni post hoc, *n* = 3. Abbreviations: DOC2A, Double C2 domain alpha; RT-qPCR, real-time quantitative PCR.

**Figure 4 ijms-23-10204-f004:**
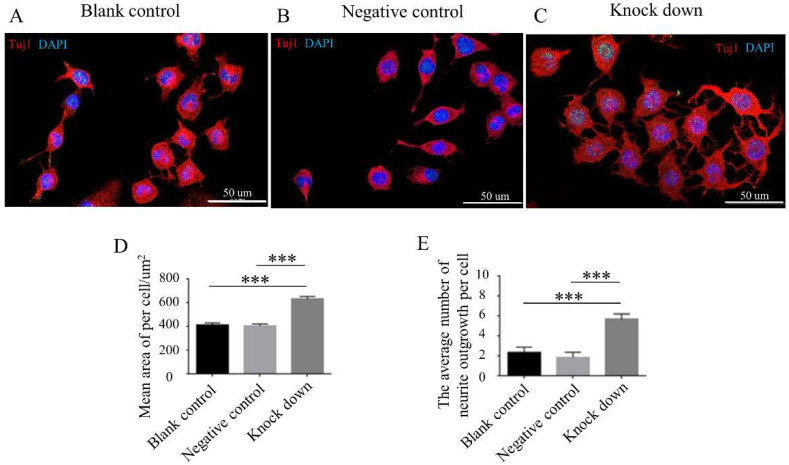
Comparison of neuron polarity among different N-2a cell groups. (**A**–**C**) Double immunofluorescent labeling (Tuj1) of N-2a cell lines. (**D**,**E**) Comparison of mean area of per cell/um^2^ and amount of neurite outgrowth per cell in different groups. ***, *p* < 0.001, one-way ANOVA with Bonferroni post hoc, *n* = 3. Knockdown, infected with *Doc2a*-lentivirus; negative control, Lentivirus Luciferase Reporter negative control; blank control, cultured with an equal medium.

**Figure 5 ijms-23-10204-f005:**
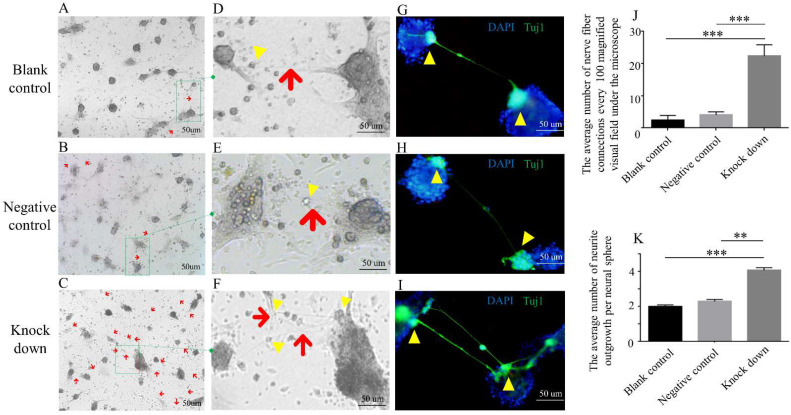
Comparison of neuron polarity among different neural sphere groups. (**A**–**C**) Nerve fiber phenotypes of neural spheres in different groups. (**D**–**F**) Partial enlarged drawing of (**A**–**C**) separately. (G–I) IF (Tuj1) showed the nerve fibers and neurites of neural spheres. Red arrow, nerve fiber; yellow arrowhead, neurite. (**J**) Comparison of average number of nerve fiber connections. (**K**) Comparison of average number of neurite outgrowth per neural sphere. **, *p* < 0.01; ***, *p* < 0.001, one-way ANOVA with Bonferroni post hoc, *n* = 3. Knockdown, infected with *Doc2a*-lentivirus; negative control, Lentivirus Luciferase Reporter negative control; blank control, cultured with an equal medium. Abbreviations: IF, immunofluorescence; DAPI, 4’,6-diamidino-2-phenylindole.

**Figure 6 ijms-23-10204-f006:**
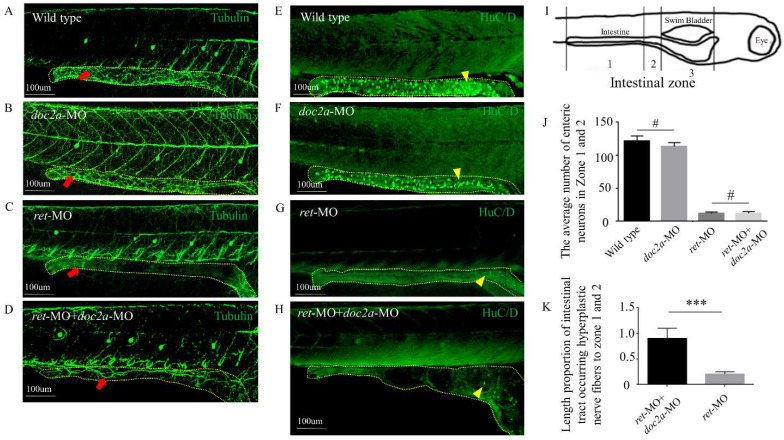
Investigation of Doc2a down-regulation promoting hyperplastic nerve fibers in zebrafish intestinal tract. (**A**–**D**) Comparison of enteric nerve fibers (Tubulin) in different groups. (**E**–**H**) Comparison of enteric neurons (HuC/D) in different groups. Red arrow, enteric nerve fiber; yellow arrowhead, enteric neuron. (**I**) Diagram of zebrafish. (**J**) Comparison of average number of enteric neurons in zone 1 and 2. (**K**) Comparison of length proportion of intestinal tract with hyperplastic nerve fibers in zone 1 and 2. #, *p* > 0.05; ***, *p* < 0.001, one-way ANOVA with Bonferroni post hoc, *n* = 3. Abbreviations: MO, Morpholino; ret, ret proto-oncogene.

**Figure 7 ijms-23-10204-f007:**
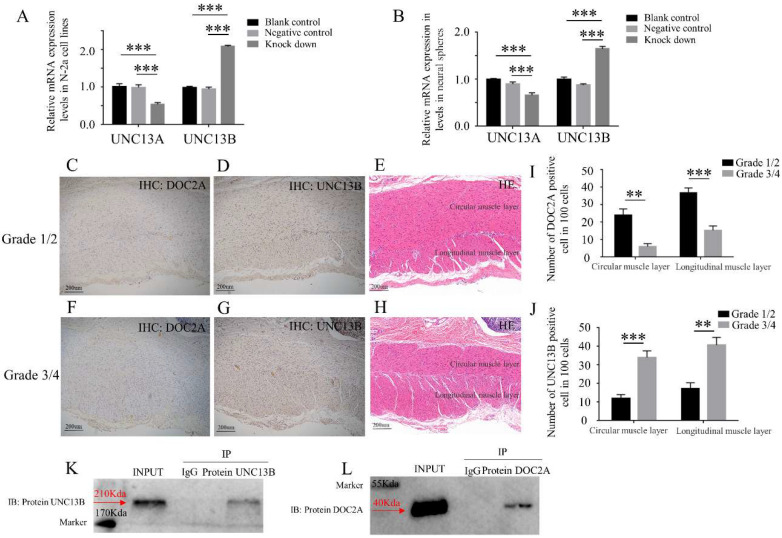
Expression relationship between DOC2A and UNC13B. (**A**,**B**) Change in UNC13B after DOC2A being knocked down in N-2a cell lines and neural spheres. (**C**–**J**) Expression levels (IHC) of DOC2A and UNC13B in different grades of HSCR patient colons. **, *p* < 0.01; ***, *p* < 0.001, one-way ANOVA with Bonferroni post hoc, *n* = 3. (**K**,**L**), protein interaction (CO-IP) between DOC2A and UNC13B. Abbreviations: UNC13B, unc-13 homolog B; IHC, immunohistochemistry; HE, hematoxylin-eosin staining.

## Data Availability

Data supporting the findings of this work are available within the paper and its Appendix A.

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
