# Peer review of "Down-Regulation of Double C2 Domain Alpha Promotes the Formation of Hyperplastic Nerve Fibers in Aganglionic Segments of Hirschsprung’s Disease"

_ijms, 2022, doi:10.3390/ijms231810204_

Round 1
Reviewer 1 Report
The authors presented that the expression of DOC2A in aganglionic segments was correlated with AChE+ grades. They found that down-regulation of DOC2A can increase the number of nerve fiber connections, similar to the phenomenon that parasympathetic and sympathetic fibers proliferate largely and distribute intricately in aganglionic segments of HSCR. However, knocking down doc2a in zebrafish did not result in loss of intestinal neurons or hyperplastic nerve fibers as a different concentration gradient of MO was injected, which was different from the phenotypes of cell experiments. We speculated that zebrafish might have a strong compensatory mechanism in response to down-regulation of DOC2A, and the function of DOC2A in the intestinal tract is not significant. They also proved that Unc13a was down-regulated while Unc13b was up-regulated. It was confirmed that there are protein interactions between DOC2A and UNC13B by co-immunoprecipitation. UNC13B seems to be a downstream molecule to DOC2A, which may participate in the spasm of aganglionic segments of HSCR patient colons. However, the signaling pathway mechanisms remain to research further.
The presented results are valuable and may solve the problem concerning children suffering from common developmental anomalies of the gastrointestinal tract.
The topic of the presented results matches the scoop of the Journal.
The experiments are well planned and carried out carefully step by step. The paper is well written.
The paper may be considered for publication after supplementing according to the following questions and comments:
1. In the manuscript Double C2 domain alpha is used as an abbreviation but spelled alternately between large (DOC2A) and small letters (doc2a). And I don't know where these differences come from. Please explain and clarify it.
2. The authors wrote: “The optimum injection concentrations of doc2a and ret were 2 ng/ul, separately. Then we set three groups to investigate the phenotypes of double MO (doc2a and ret) injection, which were ret-MO group, ret-mismatch-MO group, and ret-doc2a-MO group.”
Why did the authors use only one concentration of doc2a and ret in the performed experiments? On the basis of what studies have this concentration been determined as optimal? I recommend that these studies should be supplemented with concentration-dependent determinations.
3. The experimental part concerning the application of transfection (4.6) and Immunofluorescence (IF) staining of N-2a cells and neural spheres (4.7) are written very superficial and, in my opinion, impossible to repeat. Please provide a clearer description.
Reviewer 2 Report
Manuscript entitled „ Down-regulation of DOC2A promotes the formation of hyperplastic nerve fibers in aganglionic segments of HSCR” is an interesting, well-written and well-planned experimental work. I fully support the publication of this manuscript; however I recommend the minor revision of manuscript. Small corrections should be made to the text according to the following comments:
Title
All abbreviations used in the title of the manuscript should be explained with the full name or the full name should be left without abbreviations
Results
Figure 1 – explain for the first time in the figure abbreviation HSCR; ENS abbreviation is assigned to two different terms, please organize it;
Line 74 – should be vertebrae not vertebra
Figure 3 - explain for the first time in the figure abbreviation DOC2A, RT-qPCR
Line 105 – should be Doc2a
Figure 5 line 119 - explain for the first time in the figure abbreviation IF, explain what the abbreviation DAPI means
Figure 6 - remove the description “Knockdown, infected with Doc2a-lentivirus; negative control, Lentivirus Luciferase Reporter negative control; blank control, cultured with equal medium” from figure as it is not used in this figure; explain the full name of the abbreviations of the individual experimental groups shown in this figure
Line 158 – I think should be Unc13b not Uuc13b
Figure 7 – explain in full name all abbreviations which appear for the first time such as Unc13b, IHC, HE
Discussion
Line 198 – write the full name of abbreviation ICCs
Materials and Methods
Line 256 and 260 – write the full name of abbreviations MEM and FGF
Line 308 - write the full name of abbreviations PBS and BSA
References
All references should be changed in accordance with the rules required by the journal:
References should be described as follows, depending on the type of work:
Journal Articles:
1. Author 1, A.B.; Author 2, C.D. Title of the article. Abbreviated Journal Name Year, Volume, page range.
The citation of the literature items in the manuscript text should be ordered as there is no citation of item 23
Round 2
Reviewer 1 Report
I would like to thank the authors for their very detailed responses to the reviews' comments. The manuscript was corrected according to the suggested comments. In my opinion, the manuscript may be considered for publication in its present form.